# Multi-Task Learning Radar Transformer (MLRT): A Personal Identification and Fall Detection Network Based on IR-UWB Radar

**DOI:** 10.3390/s23125632

**Published:** 2023-06-16

**Authors:** Xikang Jiang, Lin Zhang, Lei Li

**Affiliations:** School of Artificial Intelligence, Beijing University of Posts and Telecommunications, Beijing 100876, China; jiangxikang@bupt.edu.cn (X.J.);

**Keywords:** Impulse Radio Ultra-Wideband (IR-UWB) radar, personal identification, fall detection, multi-task learning, Transformer

## Abstract

Radar-based personal identification and fall detection have received considerable attention in smart healthcare scenarios. Deep learning algorithms have been introduced to improve the performance of non-contact radar sensing applications. However, the original Transformer network is not suitable for multi-task radar-based applications to effectively extract temporal features from time-series radar signals. This article proposes the Multi-task Learning Radar Transformer (MLRT): a personal Identification and fall detection network based on IR-UWB radar. The proposed MLRT utilizes the attention mechanism of Transformer as its core to automatically extract features for personal identification and fall detection from radar time-series signals. Multi-task learning is applied to exploit the correlation between the personal identification task and the fall detection task, enhancing the performance of discrimination for both tasks. In order to suppress the impact of noise and interference, a signal processing approach is employed including DC removal and bandpass filtering, followed by clutter suppression using a RA method and Kalman filter-based trajectory estimation. An indoor radar signal dataset is generated with 11 persons under one IR-UWB radar, and the performance of MLRT is evaluated using this dataset. The measurement results show that the accuracy of MLRT improves by 8.5% and 3.6% for personal identification and fall detection, respectively, compared to state-of-the-art algorithms. The indoor radar signal dataset and the proposed MLRT source code are publicly available.

## 1. Introduction

Due to the rapidly increasing aging population and the COVID-19 pandemic, assisted living systems with intelligent personal identification have become concerning issues in IoT-based smart healthcare systems. Additionally, correctly identifying a person involved in a falling incident and providing timely warnings to nursing workers can help prevent severe injuries. Contact devices such as electrocardiogram (ECG) monitors and photoplethysmograph (PPG)-based wearable devices are commonly used for person identification. However, these devices have limitations and low adaptability in daily living conditions and movement [1,2] as they are placed close to the chest or directly on the skin. On the other hand, vision sensor-based non-contact identification methods are susceptible to lighting conditions and raise privacy concerns.

Radar fulfills these unobtrusiveness requirements as a non-contact sensor, which is privacy-preserving and able to detect both body and cardiorespiratory movements. To enable personal healthcare in an indoor environment, different radar-based methods have been researched in recent years for personal identification. Ref. [3] designed the Mono-pole UWB MIMO antenna to improve radar’s ability for short-range indoor applications. Some researchers extract micro-Doppler features using the Short-Time Fourier Transform (STFT) of human gait [4] or cardiac motility [5] and employ Deep Convolutional Neural Networks (DCNNs) for personal identification. In [6], Range-Doppler heat maps are extracted from radar signals, and classical deep learning models such as AlexNet, VGGNet, GoogLeNet, and ResNet are used for individual identification using millimeter-wave (MMW) radar. A summary of recent advances in identification based on Doppler radar systems is provided in [7]. However, these research studies often require participants to sit still or move in a specific pattern to minimize motion interference, limiting their application in daily living conditions. In our proposed method, we aim to remove signal distortions caused by person’s motions without imposing restrictions on their movements.

As for fall detection, most existing radar-based methods rely on extracting a set of features [8,9] from the radar signals and developing a supervised deep learning network [10,11,12] to distinguish between fall and non-fall daily activities. However, these artificially extracted features are highly dependent on the experimental environment, especially the recorded individuals, which affects their robustness. Recently, the Transformer network has dominated in natural language processing (NLP) and has been extended to other fields for its superior performance compared to traditional deep convolutional neural networks. The core of the Transformer is the self-attention module, which takes the sums of input embedding and positional encoding as input and maps them to produce query, key, and value matrices for each word. The attention weights between any words can be generated by dot-product query and key matrices.The weighted sum of value and attention is the attention feature. This mechanism is well suited to dealing with time-series signal as radar. Ref. [13] proposes an Transformer network named Radar Transformer which makes full use of multimodal features information of the Automotive MMW radar point cloud to realize object classification. It utilizes attention mechanism and adopts the combination of vector attention and scalar attention to make full use of the spatial information, Doppler information, and reflection intensity information of the 4-D radar point cloud to realize the deep fusion of local attention features and global attention features. Their work enlightens us to utilize Transformer in radar signal analysis to automatically explore temporal correlation features for fall detection and personal identification.

In indoor environments, it is beneficial to determine the identities of different individuals while also detecting falls to enhance personalized monitoring performance. As a result, several methods [14,15,16] have been proposed to achieve both personal identification and timely fall detection within multi-task learning architectures. However, these methods based on lidar, seismic, and camera technologies can be heavily affected by environmental lighting conditions or vibrations, thereby limiting their applicability in daily living environments. To the best of our knowledge, none of the previous works considering personal identification and fall detection have been conducted using radar sensors, which offer superior robustness compared to lidar or camera sensors.

Inspired by the Transformer and self-attention mechanisms, we propose the Multi-task Learning Radar Transformer (MLRT): a personal Identification and fall detection network based on IR-UWB radar as shown in Figure 1. The main contributions are listed as follows:

1. To deal with different persons in different environments, a multi-task learning radar Transformer network is proposed for both personal identification and fall detection to utilize the radar time-series signals. MLRT takes a “backbone-discriminator” multi-task learning network to exploit the correlation between the personal identification task and the fall detection task, thus enhancing the performance of discrimination. The proposed method is more robust and has better performance on radar signals than state-of-the-art methods.

2. Considering the signal distortions from moving persons and background noises which can affect the performance of MLRT, a signal processing method is applied on radar signals before feeding forward into the network. After Direct Current (DC) removal and bandpass filtering, a Running-Average (RA) method is applied for clutter suppression. A Kalman filter is applied for trajectory estimation.

3. A dataset is generated with an IR-UWB radar for personal identification and fall detection. The radar data are generated in the indoor environment with a total of 11 persons × 10 min. The experiments are performed in the zone in front of the UWB radar sensor, approximately 5 m × 5 m. Radar data are collected from a person who suddenly falls while walking randomly. The proposed MLRT and other existing methods are evaluated on this dataset. This dataset is now available at https://github.com/bupt-uwb/MLRT (accessed on 23 March 2023).

The rest of this paper is organized as follows. Section 2 presents the proposed MLRT method. Section 3 gives an overview of the experimental set and generated dataset. Section 4 discusses the results of the experiment. Section 5 presents the conclusions.

## 2. The Proposed MLRT Method

### 2.1. Radar Signal Model and Preprocessing

IR-UWB radar periodically transmits narrow impulse signals with wide bandwidth. The received signal can be expressed as the sum of the channel’s response and variations caused by vital signs:(1)$$r\left(t,\tau \right)=\sum_{i}a_{i}p\left(\tau -\tau_{i}\right)+a_{v}p\left(\tau -\tau_{d}\left(t\right)\right),$$
where *t* is the pulse accumulative time, τ is the pulse sampling time, p(τ) is the transmitted pulse, $a_{i}$ is the amplitude of each multipath component and $a_{v}$ is the amplitude of the vital signs. $\tau_{i}$ and $\tau_{d}$ denote the time delay in the process of signal transmission and reception, and $\tau_{d}\left(t\right)=\frac{2d_{c}\left(t\right)}{c}$.

The received radar data are stored in the form of matrix $R\left[n,m\right]$ after sampling:(2)$$R\left[n,m\right]=r\left(nT_{s},mT_{f}\right)-\frac{1}{M}\sum_{i}^{M}r\left(nT_{s},iT_{f}\right),$$
where $T_{s}$ and $T_{f}$ are the sampling intervals in slow time and fast time, respectively. Each row of matrix *R* represents the *n*-th received frame with M fast time sampling points (n=1,2,3,…,N;m=1,2,3,…,M). The signal propagation environment is static, and the movements in the environment is caused by human activities. To distinguish the static components of the radar signal from the dynamic components, the first step is to remove the average value of the signal, also known as the DC offset, from the received signal. This removes ambient static echoes that may interfere with the dynamic components of the signal.

After subtracting the DC offset, the signal is then filtered with a band-pass filter that matches the radar’s operating band, which is 6–8.5 GHz. The filter helps to remove additional noise from the signal. To extract the human body signal from raw data signal that may contain background noise and stationary clutter, the RA algorithm [17] is used. This algorithm helps to generate a clutter-suppressed signal by subtracting the estimated clutter from the received raw data signal. The estimated clutter signal can be expressed as follows:(3)$$C\left[n,m\right]=\alpha C\left[n-1,m\right]+\left(1-\alpha \right)R\left[n,m\right],$$
where $C\left[n,m\right]$ denotes the estimated clutter signal at the *n*-th slow time, and α is the gain factor which can determine the renewing ratio of the clutter signal. The radar matrix ${\hat{R}}_{N\times M}$ after preprocessing can be obtained by subtracting the estimated clutter signal.

When a person moves randomly in the room, the distance between the target and the radar is constantly changing. The target’s motion follows an integral random walk in daily living environments. Based on the radar matrix ${\hat{R}}_{N\times M}$ after preprocessing, the initial target location $Tr_{t}$ is determined by identifying the maximum energy in the range dimension of the signal matrix. Subsequently, the location estimation is updated using the Kalman filter [18] based on the minimum mean square error (MMSE) approach. Firstly, the covariance matrix of the error $P_{t}$ is determined from the following:(4)$$\begin{array}{cc} P_{t} & =P_{t-1}+Q\\ K_{t+1} & =\left(P_{t}+Q\right)/\left(P_{t}+Q+R\right)\\ Tu_{t+1} & =\left(1-K_{t+1}\right)Tu_{t}+K_{t+1}Tr_{t+1} \end{array}$$
where *Q* is a constant that affects the weight of the predicted value. *R* is the variance of the noise. The Kalman gain factor $K_{t}$ is determined by the constant *Q* and *R*. $P_{t}$ and $K_{t}$ are updated iteratively over time. The updated trajectory Tu is determined by the last estimate and the measured value, which is shown in Figure 2. The signal intensity values along the trajectory are preserved and others are set to zero in order to the remove interference.

### 2.2. Transformer Multi-Head Attention Network for Radar Signals

To enhance the performance of radar-based personal identification and fall detection, it is crucial to utilize the temporal and spatial features present in radar echo signals, namely the slow-time and fast-time components mentioned earlier. While traditional CNN-based networks excel at extracting spatial features from various data modalities such as images and remote sensing signals, they struggle to effectively capture temporal features as they are inherently time-independent. On the other hand, Recurrent Neural Networks (RNNs) and their variants, such as Long Short-Term Memory (LSTM) and Gated Recurrent Unit (GRU), are specifically designed to model and retain temporal correlations in time-series signals. However, these networks operate sequentially, which can result in information loss for long sequences and hinder parallel computation. Transformer networks, on the other hand, overcome this limitation by leveraging self-attention mechanisms, which have gained recognition for their ability to handle time-series data. However, directly applying Transformer networks to radar signals may yield suboptimal results. Therefore, it is essential to adapt the architecture of Transformer networks to suit the characteristics of radar signals and their specific applications.

The fast-time dimension, which represents the spatial dimension of the radar signal matrix, covers a wide range of distances. However, the echo signals reflected from the target person only occupy a relatively small portion of this dimension, indicating that some unnecessary or redundant features may be extracted. To address this, a CNN layer can be employed to compress information along the fast-time dimension and extract spatial features simultaneously. To preserve the time-dependent features along the slow-time dimension of radar signals, the same convolutional operations with identical parameters need to be applied to each individual fast-time slice, which corresponds to each row of the radar signal matrix. To achieve this, a Time-distributed CNN layer is proposed. This layer consists of a 1D Convolutional layer, which performs convolution calculations on each row of the radar signal matrix independently in time. Along the slow-time dimension, the parameters of the convolutional kernel remain unchanged until the entire time-series radar signal is processed. To extract effective spatial features from the radar signal, the size of the convolutional kernel should be close to or slightly smaller than the size of the target to be detected. In the experimental settings of this article, one person occupies approximately 30 columns in the radar signal matrix. Hence, a 1D Convolutional layer with a kernel size of 20 and padding of 2 is utilized in the Time-distributed CNN layer.

After the Time-distributed CNN layer and a dropout layer to address overfitting concerns, the attention mechanism from Transformer is employed to assign appropriate weights to the time-series radar signal, highlighting its temporal features. In contrast to traditional manual feature extraction methods, the attention mechanism effectively concentrates on valuable features, allowing neural networks to focus on subsets of features that are most informative. The multi-head attention mechanism is derived from the encoder module in Transformer. It captures time-related information from the features and finds widespread application in machine translation, natural language processing, and other domains. This mechanism resolves the issue of the model excessively attending to its own position when encoding information about the current position. The module is formed through the combination of several self-attention operations. The key is matrix Query (*Q*), Key (*K*) and Value (*V*). The three matrices are obtained by linear transformation through the same input. Then, the attention score is calculated as
(5)$$\mathrm{Score}\left(Q,K\right)=\mathrm{softmax}\left(\frac{QK^{T}}{\sqrt{d_{k}}}\right)$$
where $d_{k}$ is the column number of *Q*. The final output is obtained by multiplying the score matrix and *V*.

Multi-head attention can get information from different representation subspaces at different positions. Each head is similar to the Self-Attention given by
(6)$$head_{i}=\mathrm{Score}\left(Q_{i},K_{i}\right)V_{i}=\mathrm{softmax}\left(\frac{QK^{T}}{\sqrt{d_{k}}}\right)V_{i}$$

Then, the Multi-head attention concatenates all the heads and obtains the output through a linear transformation. It can be expressed as
(7)$$Multi-head\left(Q,K,V\right)=\mathrm{Concat}\left(head_{1},\ldots ,head_{n}\right)W^{o},$$
where $W^{o}$ is a weight matrix which is used for linear transformation.

Finally, after linear mapping and concating, the space and temporal features from radar signals are all extracted by the backbone network as shown in Figure 1. These features can be utilized in the subsequent discrimination tasks.

### 2.3. Multi-Task Learning-Based Personal Identification and Fall Detection

Considering that each person has a fixed activity pattern that includes walking and falling, it is natural to simultaneously address both fall detection and personal identification tasks. Multi-task learning is well-suited for handling multiple interconnected tasks simultaneously. When training a neural network with a relatively small dataset, multi-task learning maximizes the utilization of information provided by multiple labels, which helps mitigate overfitting and improves the network’s generalization ability. Additionally, multi-task learning enables data augmentation, taking into account the presence of different noises across different tasks. By leveraging the assumption that these noises tend to differ in direction, multi-task learning can reduce the impact of noise and enhance the network’s robustness.

In recent research on multi-task learning, the widely accepted approach is to use a “backbone-head” architecture. The “backbone” refers to shared layers that extract features from the input data, which are then used by the “head” to accomplish the specific goals of each task, such as regression or classification. In the case of MLRT (Multi-task Learning Radar Transformer), a similar architecture called “backbone-discriminator” is employed. This architecture is used because both personal identification and fall detection tasks involve classification, but with different numbers of classes.

To apply multi-learning on both fall detection and personal identification, a joint loss function should be designed. The joint loss function of two classification tasks can be expressed as:(8)$$loss_{whole}=loss_{1}+\lambda *loss_{2}$$
where λ is the weight factors between two tasks.

The value of λ can be affected by the relative loss values of each task and balancing the importance of two tasks. In MLRT, personal identification and fall detection are considered equally significant. Therefore, the value of λ mainly depends on relative loss values of each task. In the experimental setup of this article, personal identification is a task involving 11-class discrimination, while fall detection is a task involving 2-class discrimination. When using CrossEntropy Loss as the loss function, the relative loss value between the two tasks is approximately 1:1.15. Therefore, to achieve a balance, the parameter λ should be set to 1.15.

The whole architecture and parameters of MLRT is shown as Table 1.

## 3. Experiment Set

In the experiment, a Xethru X4M03 radar (NOVELDA Oslo Co. Ltd., Oslo, Norway) is used for data collection. Figure 3 shows the equipment and system deployment of our experiments. The IR-UWB radar operates in 6.0–8.5 GHz and has a sampling rate of 23.328 GHz, providing a high spatial resolution of 0.0067 m at a distance from 0.2 to 9.9 m. The experiments are performed in the zone in front of the radar sensor, approximately a 5 m × 5 m area. In addition, the metal lift doors and the metal tubes in the test hall produce signals from multipath reflection. The radar is placed 1.3 m above the floor level. Due to the requirement to cover the area of fall on the ground, the radar has a depression angle of 15 degrees in the vertical direction. The radar signal frames are collected at a rate of 20 frames/s. The processing terminal of the experiment is a laptop equipped with Intel i7-11800H CPU (main frequency 2.3 GHz, 16 cores), NVIDIA RTX3060 graphics card (video memory 6 GB, 192 bit width) and 16 GB memory.

The experiments include identification and fall activities running in parallel, as seen in Figure 3. Overall, experiments are performed with 11 healthy subjects including different genders and body sizes. The gender distribution is almost equal (female n = 6, male n = 5). The height of the subjects ranges from 160 to 187 cm and the weight of the subjects ranges from 51 to 85 kg. The age of the subjects ranges from 23 to 28 years old. The relevant information about the participants in the experiments is shown in Table 2.

The experiments are divided into three parts: fall events, non-fall events, and daily activities. During a 20-s fall event, the target subject freely walks for the first 10 s and then falls down perpendicular to the radar line of sight at a distance of 2 m from the radar. Each event has a duration of approximately 20 s, and the fall occurs randomly within the last 10 s of the event.

Personal identification and fall detection are performed simultaneously from the moment the target enters the experiment zone until the end of the experiment. In non-fall events, the target subject walks randomly for 20 s without falling.

Each person’s experiment is repeated 30 times, including 15 fall events and 15 non-fall events. Additionally, to further evaluate the method’s effectiveness in real scenarios, five specific daily activities are included: squatting down, sitting, turning around (the target continuously rotates), waving arms (the target raises and rotates the arms freely), and lying down. The locations and movements of the targets are not fixed during data collection, and the targets maintain slight random body movements to simulate normal human behavior. These daily living activities consist of a total of 160 samples. Each sample has a duration of 20 s, similar to the falling samples, and the activity occurs randomly within the last ten seconds of each sample.

A sliding window approach is applied to each 20-s sample with a window length of 10 s and a step size of 0.5 s. The signals within each window are preprocessed as described in Section 2 and input into the proposed MLRT network.

## 4. Result and Analysis

To demonstrate the effectiveness of the proposed MLRT in identification, comparisons with other methods are carried out on the dataset in this article. One is the LSTM network based on LIDAR [14]. A two-layer LSTM network combined with CNN is proposed to conduct both the fall detection and the personal identification. In another study using camera-based Multi-Task Convolutional Neural Network (MTCNN) [16], a MTCNN network architecture is used to conduct personal identification, object identification and unusual behaviour identification including falling. The parameters of the two above networks are slightly modified to adapt to the radar signals in dataset. Moreover, in order to verify the effectiveness of the multi-task learning, the personal identification and fall detection network from the proposed MLRT is trained and evaluated separately, which is called MLRT_PI and MLRT_FD. The Adam algorithm is selected as the optimizer and CrossEntrop Loss as the loss function for discrimination tasks. The initial learning rate is set to 1 × 10^−3^. All the methods are trained for 50 epochs equally with a batch size of 64.

The personal identification accuracy is calculated using ten-fold cross-validation in which the training set and test set are divided by a certain ratio randomly. The performances are investigated to distinguish 11 persons with different proportions of training sets which are presented in Figure 4. When 80% of the data is used for training and 20% for testing as is commonly used in machine learning based research, MLRT has the highest average accuracy which is 98.7%. When the proportion of the training set rises up to 50%, the proposed MLRT maintains the highest performance on accuracy among the tested networks. These results indicate that MLRT is able to effectively extract useful features for personal identification from radar signals, surpassing the performance of networks designed for other signal types. The personal identification network in MLRT achieves an accuracy of 93.3%, which outperforms LSTM and MTCNN but is slightly inferior to MLRT itself. This demonstrates the effectiveness of the Transformer multi-head attention network and the multi-task learning approach.

Figure 5 presents the confusion matrix obtained from the test set, which consists of the identification of 11 persons with 10 samples for each person. The results show that the accuracy of the predictions for the 11 persons remains consistent, with only Target 9 occasionally exhibiting relatively poor performance. The mean error falls within the range of 0 to 2 samples, indicating that MLRT achieves robust performance across different individuals.

Table 3 shows the metrics of fall detection with different methods. The sensitivity gives the proportion of actual positive events that are correctly identified as positives (SE=TP/(TP+FN)) and the specificity gives the proportion of actual negative events that are correctly identified as negatives (SP=TN/(FP+TN)), where true positive (TP) is an outcome where the system correctly predicts the fall class; true negative (TN) is an outcome where the system correctly predicts the non-fall class; false positive (FP) is an outcome where the system incorrectly predicts the fall class; false negative (FN) is an outcome where the system incorrectly predicts the non-fall class. 300 examples are used to calculate the average inference time. The table reveals that LSTM and MTCNN methods also achieve high accuracy values, with MTCNN exhibiting the lowest inference time due to its shallow network architecture. However, MLRT outperforms both LSTM and MTCNN in terms of accuracy, SE, and SP, even without utilizing multi-task learning. Additionally, MLRT employs a Transformer Multi-Head Attention Encoder to extract temporal features, leading to faster training compared to LSTM-based networks. This aspect proves beneficial for online learning applications, enabling parameter fine-tuning with lower latency.

## 5. Conclusions

This article introduces the MLRT method for personal identification and fall detection in indoor living environments. It addresses challenges posed by signal distortions from moving individuals and background noise through signal processing techniques. Preprocessing steps include DC removal, bandpass filtering, clutter suppression using RA, and trajectory estimation with a Kalman filter. This mitigates interference and enhances signal quality. The MLRT framework utilizes a multi-task learning radar Transformer network that handles personal identification and fall detection tasks simultaneously. It adopts a “backbone-discriminator” architecture to leverage the inherent correlation between these tasks. Experimental results using a dataset from 11 individuals in an indoor environment demonstrate outstanding performance. MLRT achieves an average personal identification accuracy of 98.7%, surpassing state-of-the-art methods. Fall detection accuracy with MLRT is 96.5%, outperforming other approaches. These results validate MLRT’s effectiveness in accurately identifying individuals and detecting falls. Future work will extend MLRT to incorporate vital signs monitoring for personalized healthcare. Additionally, research efforts will focus on recognizing human activities and gestures.

## Figures and Tables

**Figure 1 sensors-23-05632-f001:**
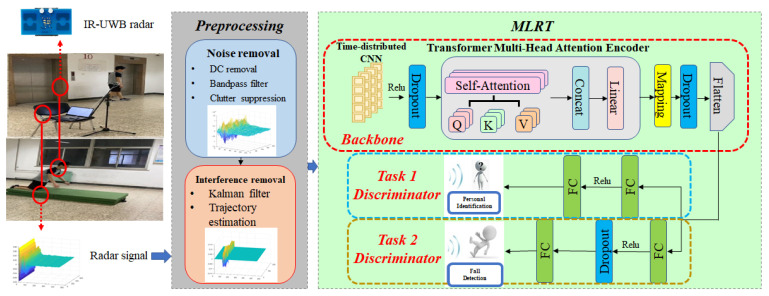
Flowchart of the MLRT.

**Figure 2 sensors-23-05632-f002:**
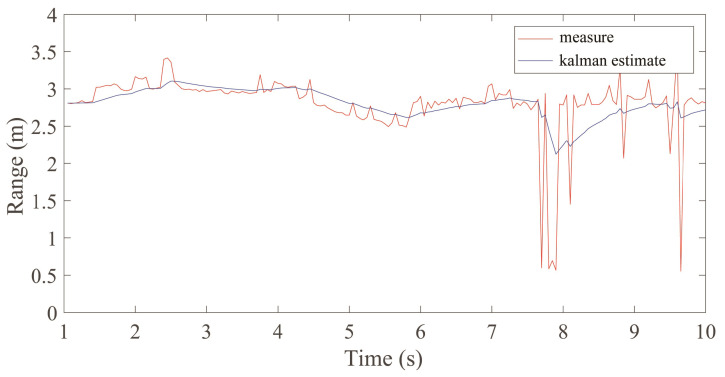
The trajectory optimization based on Kalman filter.

**Figure 3 sensors-23-05632-f003:**
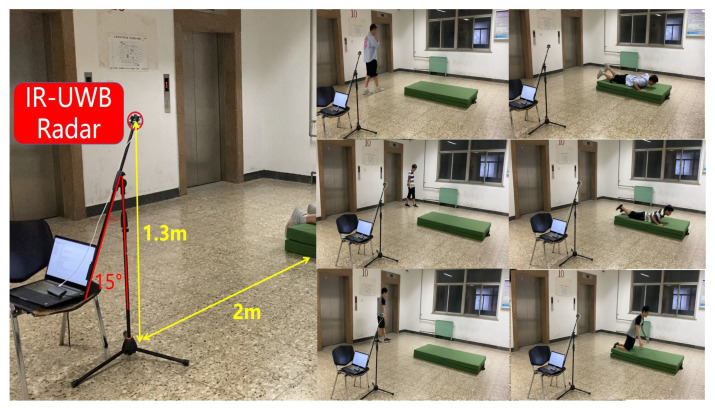
Experimental environment.

**Figure 4 sensors-23-05632-f004:**
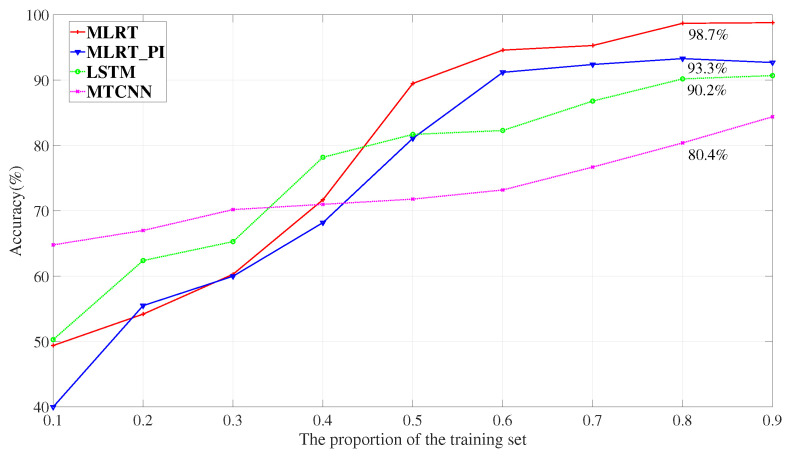
Identification results from different networks.

**Figure 5 sensors-23-05632-f005:**
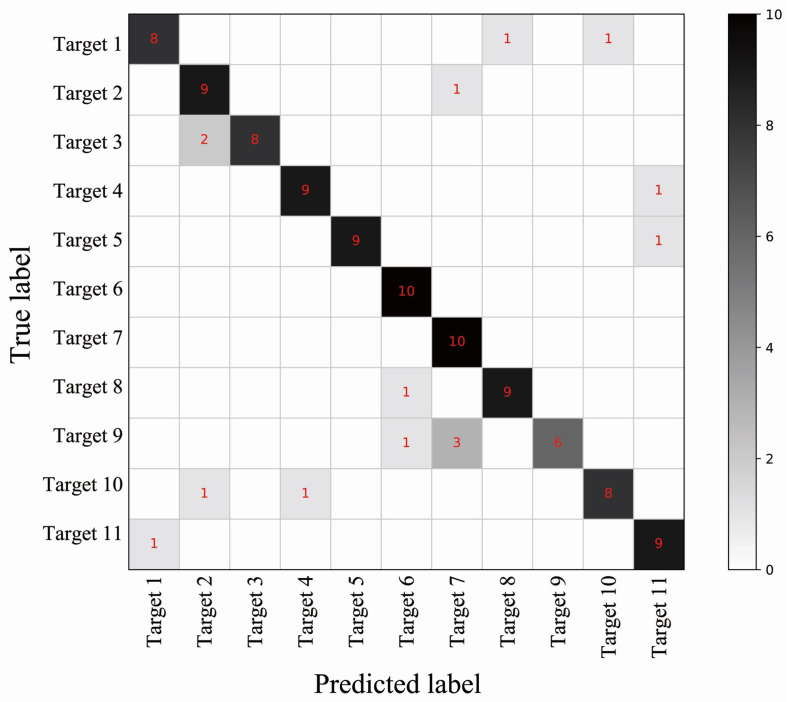
The confusion matrix for the identification.

**Table 1 sensors-23-05632-t001:** The detailed parameters of the MLRT.

Backbone	Parameters	Channel	Output Size
Input	/	1	200×1×543
Time-distributed CNN	kernel size: 20 & 1, stride: 2, ReLU	1	200×1×262
Dropout	rate: 0.2	/	200×1×262
Transformer Encoder	num head: 2	1	200×1×262
linear	output nodes: 64	200	200×1×64
Dropout	rate: 0.2	/	200×1×64
Flatten	/	/	1 × 12,800
**Personal Identification**			
FC	output nodes: 1024, ReLU	/	1024
FC	output nodes: 13	/	13
**Fall Detection**			
FC	output nodes: 1024, ReLU	/	1024
Dropout	rate: 0.2	/	1024
FC	output nodes: 2	/	2

**Table 2 sensors-23-05632-t002:** The participants in the experiments.

Targets	1	2	3	4	5	6	7	8	9	10	11
Gender	F	F	F	F	F	F	M	M	M	M	M
Age (year)	24	23	24	23	24	23	23	24	24	28	23
Height (cm)	163	171	155	160	158	167	187	170	185	174	176
Weight (kg)	54	56	50	53	52	53	83	52	72	68	76

**Table 3 sensors-23-05632-t003:** The Performance of fall detection.

	Metrics			
**Methods**	**Accuracy**	**SE**	**SP**	**Inference**
MLRT	96.5%	96.6%	96.3%	0.29 s
MLRT_FD	93.6%	95.0%	91.5%	0.30 s
LSTM	92.9%	94.3%	90.8%	0.274 s
MTCNN	90.6%	91.2%	89.6%	0.201 s

## Data Availability

The source code and radar signal dataset are available at https://github.com/bupt-uwb/MLRT (accessed on 23 March 2023).

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
