# Peer review of "Multi-Task Learning Radar Transformer (MLRT): A Personal Identification and Fall Detection Network Based on IR-UWB Radar"

_sensors, 2023, doi:10.3390/s23125632_

Round 1

Reviewer 1 Report

1. The performances are investigated to distinguish 11 persons with different proportions 266

of training sets which are presented in Figure 4. Is the dataset divided by experiments repeat or by subject, which is total different? Please make it clear.

2. Therefore [ 12 – 14 ] have been proposed to realize both personnel identification and timely fall detection in multi-task learning architectures. To the best of our known, none of previous works have been done on radar sensors which outperforms those lidar or camera sensors on robustness. 

 However, I remember that there are also some studies about the “Activity Classification and Person Identification using radar”. We think maybe you should talk about corresponding technologies and results.  

[1] A Deep Multi-task Network for Activity Classification and Person Identification with Micro-Doppler Signatures.

[2] Application of mmWave Radar Sensor for People Identification and Classification

[3] Geolocation tracking for human identification and activity recognition using radar deep transfer learning.

3. In the 20 seconds of a fall event, one target walks freely for the first 10 seconds and falls down perpendicular to the radar line of sight with a distance of 2 meters from the radar.

Each event takes about 20 seconds and the fall will occur at a random time in the last 10

seconds.

   Do volunteers always face the radar line of sight when they fall? What will happen if you don't face it right? Have you ever considered this?

4. the MLRT takes a "backbone-discriminator" multi-task learning network to exploit the correlation between personnel identification task and fall detection task to enhance the performance of discrimination.

We believe this is the main innovation point of the article. But how are the advantages of the proposed method reflected? How to prove its effectiveness? It was not clearly presented in the article

no

Author Response

  1. The performances are investigated to distinguish 11 persons with different proportions of training sets which are presented in Figure 4. Is the dataset divided by experiments repeat or by subject, which is total different? Please make it clear.

Response: Thank you for your concerns. All the dataset division is conducted randomly without manually selection in order to prove the robustness of the proposed method. We have added the description in line 268, page 8 in our article which is "The personal identification accuracy is calculated using ten-fold cross-validation in which the training set and test set are divided by a certain ratio randomly."

  1. Therefore [ 12 – 14 ] have been proposed to realize both personnel identification and timely fall detection in multi-task learning architectures. To the best of our known, none of previous works have been done on radar sensors which outperforms those lidar or camera sensors on robustness. However, I remember that there are also some studies about the “Activity Classification and Person Identification using radar”. We think maybe you should talk about corresponding technologies and results.

[1] A Deep Multi-task Network for Activity Classification and Person Identification with Micro-Doppler Signatures.

[2] Application of mmWave Radar Sensor for People Identification and Classification

[3] Geolocation tracking for human identification and activity recognition using radar deep transfer learning.

Response: Thank you for your suggestion. The 3 articles you kindly mentioned is actually for “activity recognition/classification” which is different from the target task “fall detection” in our article. We are also conducting researches into human activity recognition based on radar signals which will be introduced in our future submission. This has been added in line 322, page 10 in our revised submission which is “Additionally, the recognition of human activities and gestures will be a key focus of future research efforts.” It would be our privilege to meet with your approval.

  1. In the 20 seconds of a fall event, one target walks freely for the first 10 seconds and falls down perpendicular to the radar line of sight with a distance of 2 meters from the radar. Each event takes about 20 seconds and the fall will occur at a random time in the last 10 seconds. Do volunteers always face the radar line of sight when they fall? What will happen if you don't face it right? Have you ever considered this?

Response: Thank you for your consideration. Volunteers always stand in the center of the radar line of sight when they fall. However, their orientation is not required to be towards the radar. Theoretically speaking, when a person stands in the center of the radar line of sight and face the radar, his Radar Cross Section (RCS) reaches its maximum value in this specific distance compared to when facing the radar on his side body. In the future work, we will conduct research into the connection between human body orientation and numerical analysis of its influence on the received radar signal.

  1. the MLRT takes a "backbone-discriminator" multi-task learning network to exploit the correlation between personnel identification task and fall detection task to enhance the performance of discrimination. We believe this is the main innovation point of the article. But how are the advantages of the proposed method reflected? How to prove its effectiveness? It was not clearly presented in the article.

Response: Thank you for your concerns. The commonly used way to prove the effectiveness of multi-task learning method is to compare the evaluation outcomes with separate models for each single tasks which are trained separately without multi-task learning. We have added the corresponding experiment results in section 3 in our article to prove the effectiveness of our proposed multi-task learning method.

Specifically, in order to verify the effectiveness of the multi-task learning, the personal identification and fall detection network from the proposed MLRT is trained and evaluated separately without using multi-task learning, which is called MLRT_PI and MLRT_FD. The performance of MLRT_PI and MLRT_FD is presented in Figure4 and Table 3 separately. The accuracy of personal identification network from MLRT is 93.3%, which still outperforms LSTM and MTCNN but inferior to MLRT. This proves the effectiveness of Transformer multi-head attention network and multi-task learning method. MLRT_FD also achieves better performances on accuracy, SE and SP for fall detection task.

Reviewer 2 Report

1)      There should not be any acronyms in the title.

2)      The abstract and conclusion need to technically strengthen. Please rewrite it.

3)      In the article, it is personnel identification or personality identification. Please confirm and justify.

4)      The mathematical equations and constants mentioned must be elaborated technically in detail to understand them for physical significance.

5)      Line no 192, it is training not ‘traing’

6)      Illustrate what are the manuscript’s strengths and weaknesses;

7)      Provide some technical comments on research methods or content

Overall paper is well-written and seems technically strong after correcting these corrections/suggestions.

Author Response

  1. There should not be any acronyms in the title.

Response: Thank you for your suggestion. We make a survey on articles published in Sensors journal and find that several articles in this journal put acronym in its title listed below to name the proposed method. It would be our privilege to meet with your approval.

[1] Teršek, M.; Žust, L.; Kristan, M. eWaSR—An Embedded-Compute-Ready Maritime Obstacle Detection Network. Sensors 2023, 23, 5386. https://doi.org/10.3390/s23125386.

[2] Liang, K.; Hao, S.; Yang, Z.; Wang, J. A Multi-Global Navigation Satellite System (GNSS) Time Transfer Method with Federated Kalman Filter (FKF). Sensors 2023, 23, 5328. https://doi.org/10.3390/s23115328

[3] Erskine, S.K.; Chi, H.; Elleithy, A. SDAA: Secure Data Aggregation and Authentication Using Multiple Sinks in Cluster-Based Underwater Vehicular Wireless Sensor Network. Sensors 2023, 23, 5270. https://doi.org/10.3390/s23115270

  1. The abstract and conclusion need to technically strengthen. Please rewrite it.

Response: Thank you for your suggestion. We have rewritten the abstract and conclusion to provide a more detailed description of the specific implementation of our proposed Multi-task Learning Radar Transformer.

Abstract: Radar-based personal identification and fall detection have received considerable attention in smart healthcare scenario. Deep learning algorithms have been introduced to improve the performance of non-contact radar sensing applications. However, the original Transformer network is not suitable for multi-task radar-based applications to effectively extract temporal features from time-series radar signals. This article proposes Multi-task Learning Radar Transformer (MLRT): a personal Identification and fall detection network based on IR-UWB radar. The proposed MLRT takes the attention mechanism of Transformer as the core to automatically extract features for personal identification and fall detection from radar time-series signals. Multi-task learning is applied to exploit the correlation between personal identification task and fall detection task to enhance the performance of discrimination for both tasks.

In order to suppress the impact of noise and interference, a signal processing approach is employed including DC removal and bandpass filtering, followed by clutter suppression using a RA method and Kalman filter-based trajectory estimation. An indoor radar signal dataset is generated with 11 persons under one IR-UWB radar and evaluated the performance of MLRT. The measurement results show that the accuracy of the MLRT is improved than State-of-the-art algorithm by 8.5\% and 3.6\% on personal identification and fall detection separately. The indoor radar signal dataset and the proposed MLRT source code are public.

Conclusion: This article presents a MLRT method designed for personal identification and fall detection in indoor living environments. To address the challenges of signal distortions caused by moving individuals and background noise, a signal processing approach is employed prior to feeding the radar signals into the network. The preprocessing steps include DC removal and bandpass filtering, followed by clutter suppression using a RA method. Additionally, a Kalman filter is utilized for trajectory estimation to eliminate interference caused by human movements. By preserving the signal intensity values along the estimated trajectory and setting others to zero, the interference is effectively mitigated. The proposed MLRT framework leverages a multi-task learning radar Transformer network that simultaneously tackles personal identification and fall detection tasks. The MLRT network incorporates a “backbone-discriminator” multi-task learning architecture to exploit the inherent correlation between the two tasks. To validate the feasibility of MLRT in real-world scenarios, a dataset is collected from 11 individuals within an indoor environment. Experimental results demonstrate an average accuracy of 98.7% for personal identification, surpassing the performance of state-of-the-art methods. Similarly, the fall detection accuracy achieved by MLRT is 96.5%, outperforming others. These classification outcomes substantiate the effectiveness of MLRT in accurately identifying individuals and detecting falling events. In future work, MLRT will be further extended to incorporate vital signs monitoring, enabling personalized healthcare for multiple individuals. Additionally, the recognition of human activities and gestures will be a key focus of future research efforts.

  1. In the article, it is personnel identification or personality identification. Please confirm and justify.

Response: Thank you for your concerns. Based on the survey, we consider “personal identification” as a more appropriate name for the target task in our article, which is also frequently used in Computer Vision (CV) field. We have revised the corresponding contents in our article. Thank your again for your kind reminder.

  1. The mathematical equations and constants mentioned must be elaborated technically in detail to understand them for physical significance.

Response: Thank you for your suggestion. We have reorganized the corresponding part in Section 2 in our article to help readers better understand the basic information of our proposed method.

Specifically, We combine radar signal model and preprocessing into one subsection and only necessary formulas have been retained such as radar matrix R[n, m], estimated clutter signal C[n, m] and Kalman filter algorithm.

  1. Line no 192, it is training not ‘traing’

Response: Thank you for your kind reminder. We have fixed the typo in our article.

  1. Illustrate what are the manuscript’s strengths and weaknesses;

Response: Thank you for your concerns. The main strengths of the proposed MLRT is taking the attention mechanism of Transformer as the core to automatically extract features for personal identification and fall detection from radar time-series signals.

Multi-task learning is applied to exploit the correlation between personal identification task and fall detection task to enhance the performance of discrimination. Compared with other methods, a transformer based model is proposed which is specifically designed for IR-UWB radar signal.

However, the proposed model of our article is a data-driven method, which may have disadvantage on generalization. This can be settled to some extent by model-driven methods. A priori knowledge involving heart motion model and motion pattern recognition from radar signal analysis must be introduced if we want to apply model-driven methods to personal healthcare applications based on radar. We are currently conduct researches into this problem and are considering proposed model-driven methods for healthcare application based on different kinds of radar signals in our future work.

  1. Provide some technical comments on research methods or content.

Response: Thank you for your suggestions. We have provided technical comments to the related work in Introduction part and comparisons to our proposed method.

Specifically, for radar based personal identification and fall detection, related researches require participants to sit still to minimize motion interference or move in a designed pattern, which limit their application in daily living conditions. In our proposed method, we aim to remove signal distortions caused by motions from persons without setting restrictions to their movements.

Most of the existing radar-based fall detection methods are relied on extracting a set of features from the radar signals and developing a supervised deep learning network to distinguish between the fall and non-fall daily activities. These artificially extracted features are highly related to the experiment environments especially recorded persons, which affects its robustness. The Transformer network enlightens us to utilize Transformer in radar signal analysis to automatically explore temporal correlation features for fall detection and personal identification.

For multi-task learning that conduct personal identification and fall detection simultaneously, lidar-based, seismic-based and camera-based methods have been proposed but may receive great interference from the environmental lighting conditions or vibrations, which limits their application in daily living environments. These problems can be overcome by the usage of radar and to the best of our known none of previous works considering personal identification and fall detection have been done on radar sensors.

Reviewer 3 Report

This paper proposes a Multi-task Learning Radar Transformer to deal adar-based personnel identification and fall detection problem. The innovation is attractive. The paper is well-organized. However, the paper needs following major revisions before it can be recommended for publication:

1. In Introduction, authors should enrich the development of indoor radar systems and model-driven methods which utilizes radar Doppler information, such as

[1] Kolangiammal S, Balaji L, Mahdal M. A Compact Planar Monopole UWB MIMO Antenna for Short-Range Indoor Applications[J]. Sensors, 2023, 23(9): 4225.

[2] Zhang Y ,  Luo J ,  Li J , et al. Fast Inverse-Scattering Reconstruction for Airborne High-Squint Radar Imagery Based on Doppler Centroid Compensation[J]. IEEE Transactions on Geoscience and Remote Sensing, 2021, PP(99):1-17.

2. Compared with model-driven methods, what is the advantage and disadvantage of the proposed method?

3. How the radar parameters, for example, the bandwidth, affect the performance?

4. Author should add more experimental results for multiple persons.

5. In Result and Analysis part, model-driven methods should also be compared.

6. Abbreviations should not be redefined. For example, ‘Multi-task Learning Radar Transformer (MLRT)’ in Conclusion.

Author Response

  1. In Introduction, authors should enrich the development of indoor radar systems and model-driven methods which utilizes radar Doppler information, such as

[1] Kolangiammal S, Balaji L, Mahdal M. A Compact Planar Monopole UWB MIMO Antenna for Short-Range Indoor Applications[J]. Sensors, 2023, 23(9): 4225.

[2] Zhang Y ,  Luo J ,  Li J , et al. Fast Inverse-Scattering Reconstruction for Airborne High-Squint Radar Imagery Based on Doppler Centroid Compensation[J]. IEEE Transactions on Geoscience and Remote Sensing, 2021, PP(99):1-17.

Response: Thank you for your suggestion. The reference [1] you have recommended is relevant to our work and we have introduced it as the reference in line 32, page 1 in our article.

  1. Compared with model-driven methods, what is the advantage and disadvantage of the proposed method?

Response: Thank you for your suggestion. Model-driven methods are rather common in Computer Vision (CV) researches. However, compared to the field of computer vision, there is a limited availability of large-scale and open-source radar signal datasets that can be used for training models. Moreover, the data format from different kind of radar differs greatly, which also result in difficulties for model-driven method in learning and generalization.

Radar signal data has higher dimensionality and complex feature representations. Unlike images in CV, radar signal processing involves multidimensional information such as temporal, frequency, and amplitude aspects. This complexity poses challenges in designing and training effective models specifically tailored for radar signals. In that case, data-driven methods may have better performances especially with a relatively small dataset as the dataset we propose in our article which is captured in a specific real indoor scenario.

Data-driven methods for radar signal may have disadvantage on generalization, which can be settled to some extent by model-driven methods. However, for different AI models such as CNN, LSTM, Transformer to process radar signals, there is currently no effective interpretable method available. Radar signal processing involves various sources of physical noise and environmental interference, including multipath effects, noise contamination, and signal attenuation. These factors can impact the quality and accuracy of the signals, making model-driven methods less effective when facing these challenges. A priori knowledge involving heart motion model and motion pattern recognition from radar signal analysis must be introduced if we want to apply model-driven methods to personal healthcare applications based on radar. We are currently conduct researches into this problem and are considering proposed model-driven methods for healthcare application based on different kinds of radar signals.

  1. How the radar parameters, for example, the bandwidth, affect the performance?

Response: Thank you for your question. The X4M03 IR-UWB radar we use is produced by Novelda company and its bandwidth is set to be 1.4GHz which cannot be modified. Theoretically speaking, the range resolution increases if the bandwidth of radar increases. Therefore, the accuracy of the proposed method should also increase.

  1. Author should add more experimental results for multiple persons.

Response: Thank you for your suggestion. We mainly discuss the single person indoor environment in this article. Personal identification and fall detection for multiple persons involve multiple tracking and overlap elimination, which are what we are conducting researches into. We have mentioned works for multiple persons in the conclusion part as the future work in line 323, page 10 which is “In future work, MLRT will be further extended to incorporate vital signs monitoring, enabling personalized healthcare for multiple individuals.”

  1. In Result and Analysis part, model-driven methods should also be compared.

Response: Thank you for your suggestion. We will conduct research for model-driven methods in the future as stated in Response 2.

  1. Abbreviations should not be redefined. For example, ‘Multi-task Learning Radar Transformer (MLRT)’ in Conclusion.

Response: Thank you for your reminder. We have checked the article and removed the repeated redefinition of abbreviations.

Reviewer 4 Report

The manuscript being reviewed is the development of the author’s previous research (see Ref. [18]) in the field of Radar-based personnel identification and fall detection that have received considerable attention in smart healthcare scenario. In the manuscript, a new solution to design Multi-task Learning Radar Transformer (MLRT) that is suitable for multi-task radar-based applications to effectively extract temporal features from time-series radar signals, is proposed and verified theoretically and experimentally. In particular, an advanced multi-task learning is applied to substantially exploit the correlation between personnel identification task and fall detection task to enhance the performance of discrimination. I would like to especially note the detailed and clear description of the MRLT experimental set under study based on well-known Novelda Xethru X4M03 Impulse Radio Ultra- Wideband Radar and the quantitative comparison with the other solutions based on LIDAR or camera (see Figs. 4, 5 and Table 2). The results demonstrated that the proposed MLRT could effectively realize both personnel identification and timely fall detection in a multi-task learning architecture on radar sensors, which outperforms those lidar or camera sensors on robustness, and in the future could be combined with vital signs monitoring to further realize personal healthcare. I believe that this article has all the necessary properties in terms of significance, novelty of the solution, readability to be accepted for publication. However, its careful study showed the presence of a number of misprints. For example, line 64: “The” instead of “the”; line 72: “Kalman filter” instead of “kalman filter”; line 161: “time-series” instead of “tims-series”; line 295: “fintune” - I don't know this verb. I think “fortune” will be right; line 364: I think that the authors should give a full description of this reference link including the doi. Finally, line 324: I consider that the decision of the authors to introduce the subsection "Abbreviations" at the end of this rather difficult to read article is absolutely corrected. But its readability will be improved if it will be complete.

In general, English language is good, but there are misprints (see my previous comments)

Author Response

Thank you very much for your appreciation on our work and your efforts to revise the typos in our article. All the typos you mentioned has been revised and we want to explain specially:

1. “fintune” actually should be “fine-tune” which means modifying the parameters of the proposed AI model to improve its performance. It is our mistake to type it as "fintune". We have fixed it in line 301, page 10.

2. "Abbreviations" part has been completed to cover all the abbreviations mentioned in our article which is listed as blow:

IR-UWB    Impulse Radio Ultra-Wideband
ECG    Electrocardiogram
PPG    Photoplethysmograph
DC    Direct Current
STFT    Short-Time Fourier Transform
Millimeter Wave    MMW
RA    Running-Average
NLP    Natural Language Processing
CNN    Convolutional Neural Network
DCNN    Deep Convolutional Neural Network
RNN    Recurrent Neural Network
LSTM    Long Short-Term Memory
GRU    Gated Recurrent Unit
MTCNN    Multi-Task Convolutional Neural Network
FC    Fully Connected
MMSE    minimum mean square error

Round 2

Reviewer 1 Report

Thank you for the  response

Reviewer 2 Report

Accept in present form

-

Reviewer 3 Report

Authors answered my questions with satisfactory revisions. This paper is recommended for publication.